# Goat Milk Nutritional Quality Software-Automatized Individual Curve Model Fitting, Shape Parameters Calculation and Bayesian Flexibility Criteria Comparison

**DOI:** 10.3390/ani10091693

**Published:** 2020-09-18

**Authors:** María Gabriela Pizarro Inostroza, Francisco Javier Navas González, Vincenzo Landi, Jose Manuel León Jurado, Juan Vicente Delgado Bermejo, Javier Fernández Álvarez, María del Amparo Martínez Martínez

**Affiliations:** 1Department of Genetics, Faculty of Veterinary Sciences, University of Córdoba, 14071 Córdoba, Spain; kalufour@yahoo.es (M.G.P.I.); juanviagr218@gmail.com (J.V.D.B.); amparomartinezuco@gmail.com (M.d.A.M.M.); 2Animal Breeding Consulting, S.L., Córdoba Science and Technology Park Rabanales 21, 14071 Córdoba, Spain; 3Department of Veterinary Medicine, University of Bari “Aldo Moro”, 70010 Valenzano, Italy; vincenzo.landi@uniba.it; 4Centro Agropecuario Provincial de Córdoba, Diputación Provincial de Córdoba, 14071 Córdoba, Spain; jomalejur@yahoo.es; 5National Association of Breeders of Murciano-Granadina Goat Breed, Fuente Vaqueros, 18340 Granada, Spain; j.fernandez@caprigran.com

**Keywords:** goodness of fit, linear and nonlinear regression, mathematical modeling, parametric models, shape of milk components curve

## Abstract

**Simple Summary:**

The high costs of genotyping normally compel researchers to work with reduced sample sizes. Contextually, population observations may no longer compensate for the lack of sufficient data to fit lactation curves, hindering model efficiency, explicative ability, and predictive potential. Individualized lactation curve analyses may save these drawbacks, but may be time-demanding, which may be prevented through computational automatization. An SPSS model syntax was defined and used to evaluate the individual performance of 49 linear and non-linear models to fit the curve described by the milk components of the milk of 159 Murciano-Granadina does selected for genotyping analyses. Protein, fat, dry matter, lactose, and somatic cell counts curves were evaluated and modelled, while peak and persistence were estimated to maximize the ability to understand and anticipate productive responses in Murciano-Granadina goats, which may translate into improved profitability of goat milk as a product.

**Abstract:**

SPSS syntax was described to evaluate the individual performance of 49 linear and non-linear models to fit the milk component evolution curve of 159 Murciano-Granadina does selected for genotyping analyses. Peak and persistence for protein, fat, dry matter, lactose, and somatic cell counts were evaluated using 3107 controls (3.91 ± 2.01 average lactations/goat). Best-fit (adjusted *R*^2^) values (0.548, 0.374, 0.429, and 0.624 for protein, fat, dry matter, and lactose content, respectively) were reached by the five-parameter logarithmic model of Ali and Schaeffer (ALISCH), and for the three-parameter model of parabolic yield-density (PARYLDENS) for somatic cell counts (0.481). Cross-validation was performed using the Minimum Mean-Square Error (MMSE). Model comparison was performed using Residual Sum of Squares (RSS), Mean-Squared Prediction Error (MSPE), adjusted *R*^2^ and its standard deviation (SD), Akaike (AIC), corrected Akaike (AICc), and Bayesian information criteria (BIC). The adjusted *R*^2^ SD across individuals was around 0.2 for all models. Thirty-nine models successfully fitted the individual lactation curve for all components. Parametric and computational complexity promote variability-capturing properties, while model flexibility does not significantly (*p* > 0.05) improve the predictive and explanatory potential. Conclusively, ALISCH and PARYLDENS can be used to study goat milk composition genetic variability as trustable evaluation models to face future challenges of the goat dairy industry.

## 1. Introduction

The mathematical representation of the biological behavior of the lactation period can be very useful when breeding strategies are implemented as they permit a rather accurate control over the parameters that configure curve shape [1]. The identification of peaks and lows and increasing and decreasing sections of the lactation curve may allow adequate milking routines to the production of goats within herds. As a result, decision-making on the maintenance or elimination of females from the herd and the design of milk production simulation systems may be performed by relying on a solid basis, which may promote beneficial aspects and counteract potentially detrimental elements of the cycle [2,3]. Genetic evaluation of curve parameters may also permit the preselection of young goats, which may permit discarding low productive animals early, but also act as a potential source for identification of the underlying problems that may be causing such an underproductive status, such as subclinical pathologies [4,5,6,7]. Although the modelization of the curve of milk components has often been described for dairy cows [8], examples for dairy goats, among other species [9], especially when milk composition curves are considered at the individual level, are not frequently addressed.

Disentangling the trends described by milk components may be important, not only from an economical-productive perspective but also given the relationship of these components with a higher energy expenditure, which can condition the efficiency of management and husbandry practices. Conjoining the development of mathematical and statistical techniques with the increased accuracy that can be offered by genomic studies, we will not only be able to determine what is the current status of the animals being milked on farms, but to anticipate the future trends that their offspring will describe once they become productive. However, these studies are costly, which often compels researchers to use limited sample sizes and to design proper tools that may permit to issue valid conclusions out of such reduced sources [10,11,12].

The individualized evaluation of the goodness-of-fit of the different models available is not only important, given it enables the design of tools which better fit the real situation, but it can also prevent issues derived from incorrectly fitting biologically abnormal curves to normalized models, which may distort the real information that can be found in the field. There are multiple software possibilities available. However, their utilization requires an extra effort from researchers whose background may widely depart from biostatistics disciplines. SPSS software [13,14] improves the possibilities for the automatization of the computational stages of the process of model fitting, and at the same time it may save time resources.

Contextually, the present paper aims to describe the model syntax for the SPSS software for forty-nine models found in the literature to fit milk component curves (protein, fat, dry matter, lactose, and somatic cells). Then, to compare the model explanatory capacity and predictive ability to identify the best-fitting models in each case. Once the best-fitting functions had been determined, peak and persistency were computed for each component using the specific methods that may mathematically be more appropriate in each case. Afterwards, the model fitting properties and parameter estimation performance will be compared in a sample of Murciano-Granadina goats selected to perform genotyping studies using Bayesian methods. Conclusively, the automatized assessment of the component curves may permit the comparison of individual curves and parameters more realistically, while considering the mathematical properties of the best-fitting models for the particular situation of each animal.

## 2. Materials and Methods

### 2.1. Animal Sample and Sample Selection Process

The animals registered in the herdbook of the National Association of Breeders of Goats of the Murciano-Granadina breed (CAPRIGRAN) were ranked, depending on their official breeding value for milk yield and composition at the latest genetic evaluation at the time of sampling (published in the stud catalog in 2015). A total of 159 herdbook-registered does [15] from 28 farms in Andalusia (Spain) were considered in the analysis. All farms followed permanent stabling practices, with ad libitum water, forage, and supplemental concentrate. The Murciano-Granadina breed feed formula is standardized and a further description of the detailed and analytical composition of the diet provided to the animals in the study can be consulted in Fernández Álvarez et al. [16]. Records were collected from 2005 to 2018. The age range of the animals in the sample was 1 to 9.15 years (1.57 ± 1.11 years, mean ± SD). According to Yañez-Ruiz [17], 26.01% of Murciano-Granadina goats have their first kidding before 13–14 months.

### 2.2. Milk Performance Standardization

The Murciano-Granadina is a polyestrous breed with two kidding seasons each year, with a lactation range between 210 and 240 days [18]. The official control procedure is described in the Royal Decree-Law 368/2005 of 8th April 2005, and the milk performance recordings were performed at each farm according to the International Committee for Animal Recording (ICAR) protocol (AT4, AT4T, AT4M, A6, AT6M, or AT6T) chosen by the farmer. This legal provision suggested ICAR guidelines should be taken into consideration to perform production controls. The ICAR guidelines were set up for the first time in 1992 with a rather technically informative than normative character and have been regularly updated since then considering the new scientific computational and measuring advances to issue recommendations on the registration of diverse animal traits.

Total milk production per goat was estimated until 210 lactation days and expressed in kg as described in Pizarro et al. [19], following the protocol implemented in CAPRIGRAN (proved to be as accurate as the Fleischmann method as required in the guidelines in ICAR [20]).

The production controls considered in the present study were official, hence the normalization of real production to 210 days was performed.

To this aim, real production (*RP_j_*) for each goat was first computed as follows, $RP_{j}=d_{1}P_{1}+30\sum_{i=n}^{n_{j}-1}Pi_{j}+\left[d_{2}-30\left(n_{j}-2\right)\right]Pn_{j}$ where *RP_j_* is real production of the *jth* goat; *P*_1_ is milk yield at first control; n is the number of controls; *Pi_j_* is milk yield in ith control *i* for the *jth* goat; and *Pn_j_* is milk yield at the last control for the *jth* goat.

For the first control and the last, which were assessed individually for each goat, we computed the days (*d*_1_) between the kidding date (KD) and the date of the first control (FC), using the following formula, *d*_1_ = FC–KD; and the days between the penultimate control (PC) and the last control (LC), using *d*_2_ = LC–PC.

Afterwards, lactation yields were then standardized/normalized to provide a reasonably equitable comparison of dairy goats with different lactation characteristics, as suggested in Norman et al. [21]. The normalized milk yield per each goat at 210 days was calculated using the following formula, *NP_j_* = *d*_1_
*P*_1_ + A + *B*, where *NP_j_* is the normalized yield for goat *j*. A and *B* could be defined as $A=30\sum_{i=1}^{n_{j}-2}\frac{P_{i}P_{j}+1}{2}$, $B=\left[d_{2}-30\left(n_{j}-2\right)\right]\frac{Pn_{j}-1+Pn_{j}}{2}$.

The model used to calculate normalized yields at 210 days is described by $MP210=\sum_{i=1}^{n-1}\left[\left(\frac{pldc_{i}+pldc_{i+1}}{2}\right)\cdot \;I_{i\;i+1}\right]$, from which *MP*210 is the accumulated milk yield until 210 lactation days; *pldc_i_* is milk yield during milk control *i*; *pldc*_*i*+1_ is milk yield in the following milk control; and *I*_*i*,*i*+1_ is the day interval between two consecutive controls.

### 2.3. Milk Composition Technical Records

A number of 3107 milk component (protein, fat, dry matter, lactose, and somatic cells) controls from 399 lactations (average of 3.91 ± 2.01 lactations per goat) belonging to 159 unrelated genotyped goats were considered in the statistical analyses. Days from parity to first control were on average 21.21 ± 13.71. The number of controls per lactation was on average 4.80 ± 2.86. Primers and genotyping conditions can be consulted in Pizarro Inostroza et al. [12].

### 2.4. Milk Composition Biological Analysis and Percentual Records

Milk sampling was performed every month and analyzed at the official Milk Quality Laboratory in Cordoba (Spain). Percentages of protein, fat, dry matter, and lactose content were analyzed with a MilkoScan analyzer ™ FT1 while a Fossomatic™ FC somatic cell counter was used to test for somatic cell counts. The dataset comprised 3107 productive records for milk content (protein, fat, dry matter, lactose, and somatic cells). On-farm sampling was performed by the same operator taking milk from both mammary glands cumulatively. A minimum of two valid samples per month for each trait was necessary (milk yield and composition; protein, fat, dry matter, and lactose, except for the study of somatic cells, for which at least one valid sample per month was necessary). Samples were taken from the storage tank of raw milk and were stored and transported to the laboratory a maximum of 24 h after sampling under refrigeration conditions (between 0 °C and 4 °C or 8 °C, depending on whether conservation agents were added or not). Single samples per individual were taken from each tank. All tanks comprised raw milk on the farm at the time of collection and were marked with an individual identification label, to ensure all the necessary data to enable the analysis in the lab, to correctly identify the sample, and for sending the results to the “Letter Q database”. The date of sampling was always indicated. The Letter Q Database facilitates the traceability of milk through the identification and registration of all agents involved in the production, collection, transportation, storage, and treatment of milk and milk containers. The record number description per test is shown in Table 1.

### 2.5. Statistical Analysis

#### 2.5.1. Parametric Assumption Testing

Normality and homoscedasticity assumptions were tested on our study sample to determine whether sample properties could be biased after animal selection to configure the study sample. Shapiro-Francia (Stata Version 15.0 software, College Station, TX) was used to evaluate the normality assumption. Levene’s test (SPSS Statistics for Windows statistical program, Version 25.0, Armonk, NY) was used to test homoscedasticity. Residual values were computed after the result of the difference between the observed and predicted values. When the residuals are normally distributed, the conditional distribution of the dependent variable (Y) given the independent variables (Xs) must be normal, which means the dependent variable normally distribute at any level of the independent variables. The Shapiro-Francia test was run on the residuals of each model to determine whether they are normally distributed or not. The Durbin-Watson test [22] was conducted on the residuals of each model (using the mean percentage for each component of each day of lactation) to test for possible first-order autocorrelations among the residuals (using the linear regression test of regression procedure in SPSS version 25.0 Armonk, NY). The Durbin-Watson statistic ranges from 0 to 4. The Durbin-Watson test is reliable for sample sizes larger than 15 [23]. The Durbin-Watson statistic is only suitable for ordered time or spatial series [24], such as the ones used in the present study (days in milk).

#### 2.5.2. Composition Curve Models and Shape Parameters

One linear model and forty-eight non-linear models were used to describe the composition curves for protein, fat, dry matter, lactose, and somatic cells of the 159 does considered in the study sample. The potential differences across different lactations per animal were tested using Bayesian inference of ANOVA, as lactation order has been reported to condition milk composition. No statistically significant difference was found for any of the component across lactations (*p* > 0.05), hence we decided not to include such a factor in the models that were evaluated. Appendix A
Appendix A shows the equations for the 49 models used, the abbreviation used to identify each model, and the bibliographic references from which such information was collected. Linear and non-linear functions were used to regress the milk composition as a function of days in milk. An SPSS Model syntax for each of the 49 models in this study was described. Appendix A
Appendix A presents the SPSS equations designed to facilitate the automatized application of the models in this study. The model syntax presented is ready to be copied and pasted into the non-linear regression task from the “Regression” procedure of SPSS version 25.0, Armonk, NY [25].

The curve estimation task from the “Regression” procedure of SPSS version 25.0, Armonk, NY [25] was used to iteratively specify the parameter bounds of each model (b0, b1, b2, b3, and b4 parameters) using the Levenberg-Marquardt method of iteration [26]. The iterative process considered as many rounds as was necessary until a tolerance convergence criterion (error sum of squares between successive iterations [5]) of 10^−8^ was reached, as suggested by other authors, as stricter criteria such as 10^−6^ or 10^−8^ have been suggested to report the same outcomes out of a slightly higher number of iterations [27,28]. After convergence was reached, the initial parameters were predefined and considered to run the mechanized protocols for model fitting. A mean of 3.158 ± 0.682 (μ ± SD) iteration rounds was needed to reach the convergence criterion.

#### 2.5.3. Model Selection Criteria

The selection criteria used to determine the best explicative and predictive models included the percentage of fitted lactation curves, RSS, MSPE, adjusted R-squared (Adj. *R*^2^) and its standard deviation across the does, Akaike (AIC), corrected Akaike (AICc), and Bayesian information criteria (BIC). The Residual Sum of Squares (RSS) is a statistical technique used to measure the amount of variance in a data set that is not explained by a regression model. RSS computes the explanatory ability of the model. The cross-validated Minimum Mean-Square Residual or Error (MMSE) [29] was chosen to determine the error variation as an alternative to the cross-validated Mean-Squared Error (MSE), which has been suggested to be influenced by the number of parameters [30] if sample sizes are limited like those in genotyping studies.

In comparison to *R*^2^, adjusted R-squared or modified R-squared (Adj. *R*^2^) is a measure of the models’ ability to predict responses for new observations while it compensates (penalizes) for the overfitting event occurring after the inclusion of a high numbers of predictors. The adjusted *R*^2^ was calculated as follows: $\mathrm{Adj}.\;R^{2}=1-\left[\frac{\left(1-R^{2}\right)\left(n-1\right)}{\left(n-k-1\right)}\right]$, where *R*^2^ is the coefficient of determination, n is the number of data points, and p is the number of model parameters. The fit of a model is satisfactory if the Adj. *R*^2^ is close to one (Adj. *R*^2^ ranges from 0 to 1, with 0 meaning the model does not capture variability in the data sample and 1 meaning the model can capture all the variability in the data sampled).

Adj. *R*^2^ to *R*^2^ ratio ranges from 0 to 1 and measures the likely decrease in model fit when a certain model is applied to new data. The higher this ratio is, the less affected by overfitting the model will be (Adj. *R*^2^ should be as much close to *R*^2^ as possible for a good fit, which means overfitting may have been considered and quantified). Overfitting problems arise when the Adj. *R*^2^ to *R*^2^ ratio ranges from 0 and 0.4.

Following the premises of information theory, several methods have been described (AIC and AICc) and (BIC) of the model designed for the data being modelled. Akaike information criterion (AIC) and the corrected Akaike information criterion (AICc) were computed to compare models with regard to their ability to explain or capture the variability observed in the data set being studied and the Bayesian information criterion (BIC) was calculated to determine the predictive potential of each model, as suggested in Karangeli et al. [31].

In cases of a limited number of data points (observations) (N) or relatively complex models (high number of parameters), the corrected AICc may be more accurate. Still, AIC and AICc become similar when a higher number of observations is studied. AICc should be used when N/K < 40 [32]. As with the Adj. *R*^2^, Bayesian information criterion (BIC) is a model order selection criterion and penalizes more complicated models for the inclusion of additional parameters and was computed after Leonard and Hsu [33].

#### 2.5.4. Bayesian Model Criterion Comparison

Bayesian approximation for Pearson correlations functions was computed using the Pearson correlation task from the Bayesian statistics procedure in SPSS Statistics, Version 25.0, IBM Corp. (Armonk, NY, USA) (2017), to characterize the posterior distribution of the linear correlation among pairs of the curve shape parameters (b0, b1, b2, b3, and b4) to identify interparameter relationships. The algorithms used by SPSS for the computation of Bayesian Inference on Pearson correlation are described in IBM SPSS Statistics Algorithms version 25.0 by IBM Corp. [34]. Afterwards, the conditioning effect of model complexity on the fitting properties was tested.

Bayesian inference for ANOVA was considered given the sample size limitations and given the sample properties had violated parametric assumptions. Statistical differences in the mean for the determination coefficient (scored through Adj. *R*^2^) and flexibility selection criteria (AIC, AICc, and BIC) were tested across models consisting of two, three, four, or five regressors to determine whether model complexity can condition the best-fitting properties of the variability-capturing ability (Adj. *R*^2^), observed data explanation (AIC, AICc), and predictive potential (BIC). In this regard, smaller numerical values of the flexibility selection criteria (AIC, AICc, BIC) were reported to be indicative of better fit properties when comparing models [31].

The algorithms used by SPSS to perform Bayesian inference on analysis of variance (ANOVA) are described and publicly available in IBM SPSS Statistics Algorithms version 25.0 by IBM Corp. [34]. The tolerance value for the numerical methods and the number of method iterations was set as a default by SPSS v25.0 [25].

The estimated effect of the factors considered in the predictive models, its 95% credibility interval, and the posterior distribution statistics were computed. The significant effect of a certain factor may be detected if 0 is not contained within the credibility range.

The Bayes factor (BF) is a measure of the strength of the evidence against null or alternative hypothesis when comparing hypothesis pairs. The larger the BF the higher the evidence for the alternative hypothesis; that is, of one model over the other. Jeffreys [35] and Lee and Wagenmakers [36] set thresholds for BF to define the significance of evidence.

The Jeffrey-Zellner-Siow (JZS) mixture of g-priors [37] was used for both Bayesian inferences on Pearson’s correlations and ANOVA. Contextually, Bayes factors for JZS prior can be relatively easily and highly precisely computed [38], and have been adapted for the default *t*-test [39], ANOVA [40], and linear regression [41]. JZS prior [40] is particularly appropriate when using ANOVA as this prior is symmetric and centered at zero, in line with the predictive matching criterion as reported by Bayarri et al. [42], hence positive and negative values of the slope parameters have a priori the same probability to occur. Additionally, it is scale-invariant, hence the Bayes factors are also independent of factors or covariates. As a result, the outputs may not change if the variables measured on different units are evaluated together, which is typical in field studies metanalyses. As suggested by Rouder et al. [40], defining a scaled prior for unstandardized coefficients (β_i_) equals defining a prior for standardized coefficients $\left(\beta_{i}^{*}\right)$.

Additionally, when using the JZS prior, the scale parameter γ is fixed to a constant by the user, which allows prior beliefs to be specified about the expected effect size. The IBM Corp. [34] algorithm guide reports that the algorithm of JZS prior for linear regression analyses, to compute the Bayes factor, uses the default value of $\gamma =2\sqrt{\pi }$ = 3.5, which reflects a prior belief of a medium effect size. For a single covariate x, this choice implies that the standardized regression slope $\beta_{i}^{*}=\beta_{i}\cdot \;SD\left(x_{i}\right)/\sigma_{i}$ has an a priori probability of 53.2% of being in the range (±0.50).

#### 2.5.5. Curve Shape Parameters Computation for the Best-Fitting Model

Curve shape parameters (protein, fat, dry matter, lactose, and somatic cell count peaks and persistency values) were computed as described by the papers referenced in Appendix A
Appendix A. If the computation of peak yield was not possible, change in variable units per event was computed as suggested in Appendix A
Appendix A. Persistency and peak values should be computed differently across models as follows: the descending rate of the curve after the lactation peak, the rth relative rate of decline at the point halfway between the peak yield and end of lactation, or the instantaneous rate of change. For the cases in which no specific manner to compute the curve and shape parameters were found in the literature, the Symbolab^®^ Mathematical calculation tool for education [43] was used to determine the relative maxima (peak yield) and descending rates in the curve, depending on the model fitted (persistency).

## 3. Results

Descriptive statistics for protein, fat, dry matter, and lactose (%), as well as for somatic cell count (sc/mL) records are presented in Appendix A
Appendix A. The variation coefficient for milk components, namely, protein, fat, dry matter, lactose (%), and for somatic cell counts (sc/mL) reported a value of 14.1, 21.7, 10.0, 6.6, and 148.7%, respectively.

Appendix A show summaries of the adjusted coefficients of determination (Adj. *R*^2^), percentages of successfully fitted curves, and Adj. *R*^2^ standard deviations of the models for milk protein, fat, dry matter, lactose (%), and somatic cell counts (sc/mL) curve fitting in Murciano-Granadina goats. The adjusted *R*^2^ for the model reporting the best ability to capture variability was 0.548, 0.374, 0.429, and 0.624 for protein, fat, dry matter, and lactose content, respectively, for the model of Ali and Schaeffer (ALISCH), while the parabolic yield-density (PARYLDENS) model reported the highest values for Adj. *R*^2^ for somatic cell counts (0.481) (Figure 1).

The minimum for the Adj. *R*^2^ values (0.013, 0.029, 0.002, 0.037, and 0.000, for protein, fat, dry matter, lactose contents, and somatic cell counts) were reported for Richard’s model (RICHRDS). All goats converged for ALISCH and PARYLDENS, while the minimum fraction of goats converging for a specific model was 5.88%, 1.96%, 3.92%, and 1.96%% for protein, fat, dry matter, and somatic cell count when RICHRDS was considered. The minimum percentage of successfully fitted curves for lactose was 1.96% when modelled using the Log Modified Weibull (LGMWEIB) model. Standard deviation values for the Adj. *R*^2^ was in the range of 0.200 to 0.250 for all models (Appendix A
Appendix A). However, these values reduced in those models for which a lower number of animals successfully fitted, for instance for Gompertz (GMPRTZ), Richards (RICHRDS), and third-order Legendre orthogonal polynomial (3ORDLEG), as shown in Appendix A
Appendix A.

Parametric assumptions (normality, Shapiro-Francia test *p* < 0.05) and homoscedasticity, Levene’s test, *p* < 0.05 across groups) were violated in our study dataset, hence we opted for the use of a nonparametric statistical alternative. As the sample used in this study was small, Bayesian analyses were run in an attempt to preserve the model accuracy and power of the techniques applied.

Additionally, the Shapiro-Francia test was performed to test for the residuals’ normality, reporting statistically significant results for all fitted models (*p* < 0.001). Thus, residuals were not normally distributed. The Durbin-Watson statistic showed that all values were within the range of 0 to 2; thus, the residuals of all models were positively autocorrelated. The run test in our study indicated that the residuals of all models were not independent. These results are consistent with the earlier studies reported by Mohanty et al. [44].

Appendix A show a summary of the model curve shape parameters (b0, b1, b2, b3, and b4), the number of model regressors, measures for model fit and flexibility selection criteria computed through the Residual Sum of Squares (RSS), Mean-Squared Prediction Error (MSPE), variability explanation power through the Akaike Information Criterion (AIC) and corrected Akaike Information Criterion (AICc), and predictive power through the Bayesian Information Criterion of the models that were used to fit the Murciano-Granadina lactation curves. There was wide variability with regard to curve shape parameters. Almost all models reported values for b0 around 4, 5, 14, 5, and 800 for protein, fat, dry matter, lactose contents and somatic cell count, respectively, as shown in Appendix A, except for those implying a higher computational complexity, which in fact may have conditioned their better explicative and predictive potential (CEXPGR, CUBSPL, CURVES, DJKSTR, HAYSHI, INVQPOL, LGMWEIB, MILKBOT, NELDER, PARYLDENS, RATCUB, and 3ORDLEG for protein, fat, dry matter, and lactose; and LOGLOG, MICHMEN, MILKBOT, MORMFLO, PEMSIK, POWER, QUADSPL, RATCUB, RICHRDS, SIMLIN, SIN&GOP, VBRTLNFY, PARSURW, and WILMINK for somatic cell count).

Concretely, DENSITY, GAUSS, INVQPOL, LOGLOG, LGMWEIB, MICHMEN, PARYLDENS, 3ORDLEG, and PARSURW failed to converge for almost all milk components, hence no Adj. *R*^2^ is reported for them. The correlations between estimates of curve shape parameters (b0, b1, b2, b3, and b4) are presented in Appendix A
Appendix A. Large correlations between b0, b3, and b4 were reported (Appendix A
Appendix A) for lactose and dry matter. For somatic cell count, moderate to large negative correlations were found between b1, b4, b2, and b3. For fat and protein contents, large positive correlations were also found between b3 and b4.

Appendix A report a summary of the Bayesian ANOVA to test for differences in the mean for the adjusted *R*^2^, AIC, AICc, and BIC across models comprising two, three, four, or five regressors, respectively. Significant differences were found for the mean of the adjusted determination coefficient while no significant difference was found for the flexibility selection criteria (AIC, AICc, and BIC) when models comprised two, three, four, or five regressors. An increasing trend was described with each element added to the model (mean increase for all components was 0.111 when model complexity increased from 2 to 3 parameters, of 0.036 from 3 to 4 parameters and of 0.105 from 4 to 5 parameters, respectively). Regarding the flexibility selection criteria, the explicative and predictive potential of the models did not significantly increase nor decrease with the number of regressors considered in the models.

ALISCH was the best model, not only concerning its ability to capture population variability in all components, except for the somatic cell count for which the parabolic yield-density (PARYLDENS) resulted in the best-fitting model. This better performance could be attributed to the inclusion of logarithms, powers, and exponents or sine and cosine elements in the model, as it was also reported for other models tested such as Cubic (CUBIC), Cubic Spline function with one knot (CUBSPL), Quadratic cum log model (QDCMLOG), Grossman (GROSMN), Dijkstra (DJKSTR), Quadratic model (QUADRT), Quadratic model Dave (DAVE), Quadratic spline function with one knot (QUADSPL), Dhanoa (DHANOA), Parabolic exponential model and Parabolic, Sikka (PEMSIK), Ratio Cubics/ Partial Fraction with Cubic Denominator (RATCUB), Singh And Gopal (SIN&GOP), Cappio Borlino/biexponential (CAPBOR), Asymptotic Regression, Lactation modification of Metcherlich Law of Diminishing Returns or Exponential growth model (METLAW), Morgan Mercer Florin (MORMFLO), Wilmink’s exponential (WILMINK), and Wood (WOOD), which reported very close adjusted determination coefficient around 0.5, but presented a close value of flexibility selection criteria. These best-fitting models always included over three regressors as Table 2 had suggested being the best performing models on average regarding their ability to capture data variability; that is, explained variation. Once the best-fitting models have been identified, the summary of the results for the specific computation of peak and persistency following the methods proposed in Appendix A
Appendix A are shown in Appendix A
Appendix A.

## 4. Discussion

Models capable of forecasting the evolution of milk components and somatic cell counts provide useful information for time-related management decisions in dairy farm breeding and management programs. The highest adjusted *R*^2^ values were reported in the ALISCH model for protein, fat, dry matter, and lactose (0.548, 0.374, 0.429, and 0.624, respectively), and the PARYLDENS (0.481) model for somatic cell count. This suggests the ALISCH model enables capturing a greater fraction of the variability in the data sample when compared to other models, and the great repercussion that time evolution has on the components being measured. González-Peña et al. [45] and Harder et al. [46] found slightly lower results for the Adj. *R*^2^ of the ALISCH model, which can be attributed to the properties and characteristics of the sample that was used.

The polynomial regression of ALISCH has been reported to perform well when fitting for milk content in breeds and crossbred goats across the world. For instance, Oravcová and Margetín [47] reported *R*^2^ values that were slightly above 0.6 for fat content and above 0.7 for protein content. In this context, ALISCH has been reported to be preferable over other commonly fitted models such as WOOD or WILMINK, although it is more demanding than the aforementioned models in terms of the minimum number of test-day records required per lactation for the fitting procedures to be successful [48]. In line with these results, Buttchereit et al. [49] suggested random regression models to be superior compared with fixed regression models and, in general, the ALISCH function to be most suitable for modeling both the fixed and random regression part of the mixed model, when a sufficient number of observations is present [48], although as our results suggest this minimum limit may be somehow flexible if observations are evenly (or almost evenly) distributed across time.

In the context of studies seeking the understanding of the genetic background behind the lactation curve shape parameters, Santos et al. [50] suggested the models using the ALISCH function for the additive genetic and permanent environmental effects can be adopted, as little variation is observed in the genetic parameter estimates compared to those estimated by models using the B-spline function. Again, Santos et al. [50] reported ALISCH usually outperform other models, such as WOOD, especially in different scenarios of data distribution [51], although it tends to generate mathematical artifacts such as negative or too high predicted values of milk production at the beginning or at the end of lactation [51].

Aforementioned failures to fit specific stages of the lactation curve and its evolution should be analyzed from different statistical perspectives. First, less records are usually available at the edges of the lactation compared to middle stages; hence, shortcomings of extrapolating relationships between traits (either yield or composition) and days in milk (DIM) beyond their known range of validity may occur when models are fitted [52].

Grossman and Koops [53] also addressed an additional theoretical issue, common to several models used to fit lactation curves, which is the fact that the whole lactation is often considered as a single process. These authors introduced the assumption that lactation may result after the sum of two different overlapping phases. The diphasic model (two linear and two quadratic logistic functions) has estimated theoretical durations of the two overlapping phases of approximately 200 and 410 days. As a result, multiphasic models are characterized by a large number of parameters (three for each phase), thus requiring a greater number of tests to achieve a convenient degree of fitness, as in the case of average curves and extended lactations [5], which have been reported to frequently occur in Murciano-Granadina goats [18]. This may be supported as well on the lower values for RSS and MSPE (Appendix A), which may evidence the increased accuracy of the ALISCH and PARYLDENS models when fitting for protein, fat, dry matter, lactose, and somatic cell count, respectively, in comparison to the rest of models that were tested.

For somatic cell count, an indicator of mastitis, which is markedly influenced by other factors such as the breed being evaluated and days in milk, a slightly lower result was found. The highest somatic cells in milk were 9,756,000 sc/mL, which means some of the goats considered may potentially have had subclinical mastitis, which may have affected milk yield and composition. Additionally, the inverse proportion between somatic cells and milk yield over time may be due to the dilution factor, which means the same number of cells will give a lower cell count if the milk volume is higher [54,55].

To explain this, Bohmanova et al. [56] reported the parabolic shape of the variance function may rather be a mathematical artifact than based on a factual biological background. Still, the provided test-day records should evenly be distributed across lactation (which is an assumption to fulfil when evaluating time series-dependent data) and the overestimation and underestimation of the variances should balance across lactation; thus, it does not condition genetic evaluation of these traits.

According to González-Peña et al. [45], the Ali and Schaeffer model (ALISCH) [57] and the third-order orthogonal polynomials of Legendre (3ORDLEG) were able to recognize 9 and 14 types of curves. The correlations between the estimated parameter values for ALISCH were greater than those estimated for 3ORDLEG. These two functions have been used to model dry matter intake [58] and to estimate the genetic effect on dairy traits [59].

In the context of our results, the different shapes of the curves found when the five-parameter models were fitted derive from the specific deformation of two basic shapes (typical or atypical). Then a certain degree of variability occurs depending on the presence of inflection points in the different groups of the curves, which causes the standard individual patterns to provide an inverted form, with a phase of initial decrease to a minimum followed by an increase stage, which is common for fat and protein contents in time [60,61]

The most common shape found in the field (about 20–30% of cases) is the atypical form [8,62], which represents mainly a computational problem due to the interaction between the mathematical structure of the model used and the combinations of test day values and their distribution along the lactation trajectory [8,60,61].

According to Palacios Espinosa et al. [63] and Buttchereit et al. [49], the Ali and Schaeffer (ALISCH) and Legendre models presented better-fitting properties than Wilmink’s exponential (WILMINK) model for fat, protein, and dry matter. These better-fitting properties may base on the better adjustment to the distribution patterns, indicating that the models with fewer parameters find more difficulty to model the variation occurring along the entire curve. This suggests not only the logarithmic shapes included in computational methods promote model fit, but also exponential shapes, as it was supported by our results for Adj. *R*^2^ and Adj. *R*^2^ SD with the models of Cubic (CUBIC), Cubic Spline function with one knot (CUBSPL), Quadratic cum log model (QDCMLOG), Grossman (GROSMN), Dijkstra (DJKSTR), Quadratic model (QUADRT), Quadratic model Dave (DAVE), Quadratic spline function with one knot (QUADSPL), Dhanoa (DHANOA), Parabolic exponential model and Parabolic, Sikka (PEMSIK), Ratio Cubics/Partial Fraction with Cubic Denominator (RATCUB), Singh And Gopal (SIN&GOP), Cappio Borlino/biexponential (CAPBOR), Asymptotic Regression, Lactation modification of Metcherlich Law of Diminishing Returns or Exponential growth model (METLAW), Morgan Mercer Florin (MORMFLO), Wilmink’s exponential (WILMINK), and Wood (WOOD).

According to Palacios Espinosa et al. [63], the parameters related to the fitting performance reported by the Legendre polynomials were lower than those obtained for the Ali and Schaeffer (ALISCH) function. This behavior was referred to by Ali and Schaeffer [57], who found it difficult to find biological reasons why the functions of the Ali-Schaeffer and Legendre polynomials can detect dissimilar types of curves; however, this behavior is particularly useful for random regression analyses, where individual deviations from a curve are sought.

Assuming that the ALISCH model is a complex parametric model, given the number of elements that it comprises, our results suggest that the inclusion of logarithmic and exponential forms in the formula may somehow promote the adaptation of the curves described by the components in the milk produced by each goat individually to the properties of the model, which may result in the improvement of its ability to capture the variability of the models compared to the rest. However, Meyer [64] reported that these better model adjustment results can be balanced by the increased power of prediction or a decrease in explicative error, since random regression models using cubic, quadratic, or even higher polynomials provide erratic and unlikely estimates of variance components. The same authors may suggest this situation accentuates in contexts in which few records per animal are available in favor of third-order polynomials. However, this may contradict our findings. This result disagreement may be based upon the fact that in comparison to *R*^2^, Adj. *R*^2^ compensates (penalizes) for the overfitting event occurring after the inclusion of high numbers of predictors.

Alternatives to reduce the degree of polynomials, such as spline functions, which have also been called segmented polynomials, have also been studied. These functions are curves consisting of individual segments of low-grade polynomials that merge at specific points, called knots. Spline functions can be modeled in different ways and, depending on the choice, reduce multicollinearity, are easy to estimate, and have a simple biological interpretation.

Cubic spline regressions have been suggested to perform well given their ability to balance adjustment performance, data sensitivity, smoothness, and parameterization in average adjustment curves [65,66]. Besides, they may adapt to sudden local variations as it has been suggested for the specific case of the patterns described by somatic cell counts around a clinical mastitis event [67,68]. A technical problem in spline adjustment is represented by optimizing the number and location of the knots. Some authors recommend that knots be as many as possible, placed at maximum concentration points of records [65,66], even if such a criterion necessarily increases the number of records and function parameters.

In several documents, the number and position of the knot are fixed a priori, usually evenly spaced [65,66,69]. However, our results showed for spline functions (QUADSPL and CUBSPL with a single knowledge), slightly lower values for the ability to capture variability (Adj. *R*^2^) did not exceed those reported by the ALISCH model, for milk components, or PARYLDENS for somatic cell counts. Therefore, since the flexibility selection criteria (AIC, AICc, and BIC) were negligibly lower and statistically insignificant for CUBSPL than for ALISCH or QUADSPL, respectively, the ALISCH and PARYLDENS model remained preferable when individualized lactation curves are made.

Slight increases in the b0 shape parameter did not imply high increases in the values of the flexibility selection criteria. However, when the values for b0 highly differed from 0 in absolute value, a higher poorer ability to explain and predict was suggested, as shown in Appendix A. With only a few exceptions, values for b1, b2, b3, and b4 were maintained around 0 except for the models that were reported above to have highly increased or highly decreased values of b0. The correlation values obtained in our analyses may be supported by those of Macciotta et al. [8], as moderate to large correlations between b0, b3, and b4 were reported for lactose and dry matter, moderate to large negative correlations were found between b1 and b4 as well as b2 and b3 for somatic cell counts and large positive correlations were also found between b3 and b4 for fat and protein contents. Where b0 is responsible for the upward phase of the curve (peak) in the models per individual goat, finding ALISCH values ranging between 2.570 and 14.972 for fat (%), protein (%), dry matter (%), and lactose (%), and 717.39 for the PARYLDENS model for somatic cells (sc/mL).

In this context, our findings for those b parameters linked to the persistence are similar to those found by other authors [70,71,72] in Alpine and Saanen goats under different environmental conditions. Superior values for individual lactation curves were also found in the Ali and Schaeffer model (ALISCH), Wilmink’s exponential (WILMINK), and Wood (WOOD) models, as described by other authors [8,65,73].

Even in the context of a limited sample, our data may be in the limit to obtain reliable results, as supported by Græsbøll et al. [74], whose preliminary results, derived from a small data sample, suggested that when data is enough, there is a greater likelihood of the b parameters of the curve shape to distribute normally around a positive value different from 0, which occurred for the models fitted in our study.

Wilmink’s combined exponential (WILMINK) and linear model and the Ali and Schaeffer (ALISCH) polynomial regression can be considered as a transition between early and more recent models. The Wilmink’s (WILMINK) function consists of three terms that additively combine, which could have been presumed to improve the flexibility and variability-capturing ability performance. In our case, a significant flexibility improvement was not detected, while a strong statistical significance was reported for variability explanatory ability.

These models can be easily linearized by setting the k parameter to an appropriate fixed value [75] and its parameters still maintain a relationship to the shape of the lactation curve. The ALISCH model comprises a higher number of parameters that may allow us to adjust a wider range of shapes, even if these parameters do not have a technical meaning. Both models have been successfully used to adjust individual curves [8,65] and were implemented in previous applications of random regression models [66].

Although these models generally surpass Wood’s (WOOD) function, especially when different data distribution scenarios are compared [65]. These two models (WILMINK and WOOD) tend to produce mathematical artifacts, such as negative or too high dairy performance values predicted at the beginning or end of lactation [8,66].

Adjusting *R*^2^, via Adj. *R*^2^, and the analysis of flexibility selection criteria (AIC, AICc, and BIC) can actively help us reduce the effects of overcompensation from the inclusion of a greater number of parameters. Contextually, the penalty computed by BIC may be stricter than for the AIC for a reasonable sample size. However, for a small n, a corrected version of the AICc might be the most suitable as it would provide us with a stronger penalty than the AIC and BIC (Brewer et al. [76]).

According to Burnham and Anderson [77], the effect of a higher penalty increases the probability of the selection of smaller models to present better-fitting properties, hence the AICc tends to choose smaller models than AIC in situations of small sample sizes. As a result, BIC, in realistic situations, can tend to select models that are too simple (which may not fit). In this context, our results suggest the relationship between the complexity and flexibility of the selection criteria may not be strictly linear, while it may depend rather on the concepts of computational and parametric complexity. In this regard, despite models presenting a higher number of elements normally reporting best-fitting properties, in cases in which models presented a slightly lower number of parameters, the slight decrease in explanatory ability was compensated by the inclusion of logarithmic or exponential elements, which increased the model’s computational complexity. This computational complexity compensation made the Adj. *R*^2^ values to overcome those from the rather parametrically more complex models, as could have been expected.

Our results suggest AIC, AICc, and BIC should not be the preferable option to consider when choosing the model to describe milk component trends, but Adj. *R*^2^. This could be attributed to the individualized adjustment of the curve model, given the specific treatment of the data belonging to each specific animal may mean that the explanation of intraindividual variability is maximized as much as possible in the context of the observations available to that particular animal, but this is detrimental for other aspects, for instance, predictive potential of general models applied at a large scale.

## 5. Conclusions

Conclusively, our results emphasize that even in limited sample size contexts, the ALISCH model’s superiority in the modelization of components (protein, fat, dry matter, and lactose) may rather be ascribed to its higher computational complexity than to its parametric complexity. Parametric complexity may condition the variability capturing ability of models, but it may not alter or condition the better performance of models regarding flexibility criteria (AIC, AICc, and BIC). The PARYLDENS model’s more successful results when fitting for total somatic cell count may be attributed to the fact that it may better represent the punctual events of mastitis, which may presumably be based on the smaller number of parameters of the PARYLDENS model. Contextually, while a three-parameter model requires more records for a credible estimate, it may be able to more accurately predict persistence, given the independence of parameters. The PARYLDENS model may outperform the mean values for the Adj. *R*^2^ for models with the same or even higher number of parameters. Hence, its better explicative ability may rather be based on its computational complexity, derived from the inclusion of a negative exponent. The methodology and model syntax used here may be of use when aiming to determine the specific approach to follow internationally in an attempt to adapt to all of the wide scope of potential conditions that goat breeds may face worldwide, and which may condition the nutritional quality of the milk that they produce.

## Figures and Tables

**Figure 1 animals-10-01693-f001:**
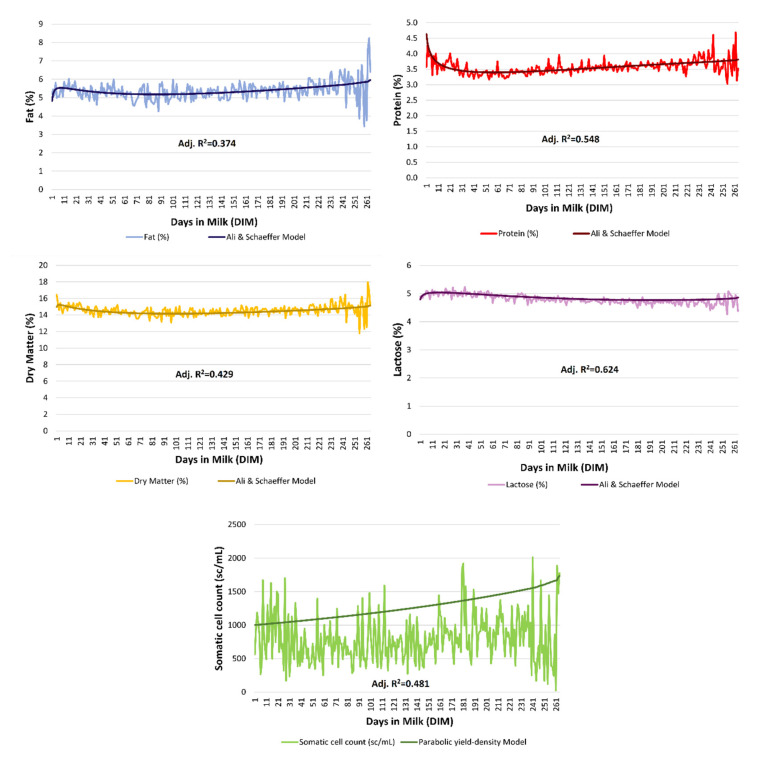
Graphic representation of best-fitting models for milk composition traits in Murciano-Granadina goats; Ali and Schaeffer model (ALISCH) for fat (%), protein (%), dry Matter (%), and lactose (%) and the parabolic yield density model (PARYLDENS) for somatic cell count (sc/mL), respectively.

**Table 1 animals-10-01693-t001:** Record number description.

Test System	Trait	Total Biological Samples Considered	Total Technical Records Considered
MilkoScan analyzer ™ FT1	Protein (%)	3107	15,535
	Fat (%)		
	Dry Matter (%)		
	Lactose (%)		
Fossomatic™ FC Somatic cell counting	Somatic cell count (sc/mL)		

The samples belonging to animals with missing or incomplete phenotype registries were discarded, hence the final set comprised observations from 159 studbook-registered goats out of the 200 animals that were initially considered.

**Table 2 animals-10-01693-t002:** Summary of the Bayesian ANOVA to test for differences in the mean for the adjusted *R*^2^ across models comprising two, three, four, or five elements.

Parameter	Protein (%)	Fat (%)	Dry Matter (%)	Lactose (%)	Somatic Cell Count (sc/mL)
Sum of Squares	0.237	0.128	0.164	0.234	0.184
df	3	3	3	3	3
Mean Square	0.079	0.043	0.055	0.078	0.061
F	4.78	6.102	4.479	3.361	4.461
Sig.	0.007	0.002	0.009	0.029	0.009
Bayes Factor	2.173	8.367	1.536	0.451	1.564
2 elements models Posterior Mean	0.215	0.130	0.153	0.312	0.136
2 elements model 95CI	0.128–0.302	0.074–0.187	0.079–0.228	0.209–0.415	0.056–0.215
3 elements models Posterior Mean	0.342	0.217	0.241	0.454	0.259
3 elements model 95CI	0.279–0.406	0.176–0.258	0.188–0.294	0.379–0.529	0.200–0.319
4 elements models Posterior Mean	0.389	0.242	0.289	0.479	0.277
4 elements model 95CI	0.267–0.432	0.169–0.271	0.195–0.330	0.333–0.529	0.174–0.325
5 elements models Posterior Mean	0.497	0.341	0.391	0.588	0.381
5 elements model 95CI	0.366–0.627	0.257–0.426	0.279–0.503	0.433–0.742	0.262–0.500

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
