# Peer review of "Goat Milk Nutritional Quality Software-Automatized Individual Curve Model Fitting, Shape Parameters Calculation and Bayesian Flexibility Criteria Comparison"

_animals, 2020, doi:10.3390/ani10091693_

Round 1

Reviewer 1 Report

The paper is of interest to the scientific fields. Correlations can be established between the multitude of constituents and the parameters in the quality assessment. 

Author Response

Reviewer 1

The paper is of interest to the scientific fields. Correlations can be established between the multitude of constituents and the parameters in the quality assessment. 

Response: We thank for the reviewer comment.

Reviewer 2 Report

My assessment of the work under the topic: Goat milk nutritional quality software automatized individual curve model fitting, shape parameters calculation and bayesian flexibility criteria comparison. 

  • In the abstract, the avearge number of lactations was 3.87 +/-2.05 lactation/goat. Check please, the values of the deviation is very high 2,05?
  • There are many abbreviations in the abstract such as (Adj. R2, MMSE, RSS, MSPE, SD, AICc, Bic, Adj. R SD), that have to be written compeletly, as many researchers only read the abstract.
  • In line 154, it was written that the age of the goat ranged between 1 and 9, 15 years. A lactation season is difficult to have at first year of age and usually after a year of age. 
  • In the table, line 332 showed that the highest somatic cells in milk was 9756000 cells/ml milk. That means that some goats had subclinical mastitis. This disease affect the milk yield and the ingredients. The question is whether or not such goats were taken into account during the study. However, line 433 and 437 were not clearly mentioned in the discussion.
  • The Tables: S1 to S11 are not in the article. They are mentioned in the supplementary materials. They are only available online. Is it compulsory to see such information only online?  

Author Response

Reviewer 2

My assessment of the work under the topic: Goat milk nutritional quality software automatized individual curve model fitting, shape parameters calculation and bayesian flexibility criteria comparison. 

  • In the abstract, the avearge number of lactations was 3.87 +/-2.05 lactation/goat. Check please, the values of the deviation is very high 2,05?

Response: There was a typo in the numbers of average lactation 3.91 ± 2.01 lactations per goat, still numbers are high as minimum lactation was one and maximum was 9.

  • There are many abbreviations in the abstract such as (Adj. R2, MMSE, RSS, MSPE, SD, AICc, Bic, Adj. R SD), that have to be written compeletly, as many researchers only read the abstract.

Response: Abbreviations were defined and abstract readapted to fit the 200 words limit.

  • In line 154, it was written that the age of the goat ranged between 1 and 9, 15 years. A lactation season is difficult to have at first year of age and usually after a year of age. 

Response: According to Yañez-Ruiz [1], 26.01% of Murciano-Granadina goats have their first kidding before 13-14 months.

  • In the table, line 332 showed that the highest somatic cells in milk was 9756000 cells/ml milk. That means that some goats had subclinical mastitis. This disease affect the milk yield and the ingredients. The question is whether or not such goats were taken into account during the study. However, line 433 and 437 were not clearly mentioned in the discussion.

Response: Information was added to the section proposed by the reviewer.

  • The Tables: S1 to S11 are not in the article. They are mentioned in the supplementary materials. They are only available online. Is it compulsory to see such information only online?  

Response: The Supplementary Tables are cited in the text of the article in:

Table S1. Line 210 and 305.

Table S2. Line 214.

Table S3. Line 307.

Tables S4-S8. Lines 370, 377 and 389

Table S9-S11. Line 399

Table S12. Line 427.

These are necessary as indeed they provide the evidence on which to support discussion and after which to draw conclusions. Tables from 1 to 5 were moved to supplementary as well as suggested by another reviewer.

Reviewer 3 Report

The introduction is unnecessarily long. Readers of the manuscript would understand the topic, hence a two-paragraph intro would suffice.

2.1. Some measures of the age of animals are important: mean±s.e. or median values will help.
2.2. polyestric, please use correct terminology and also English-english language.
2.2. kgs is wrong
2.2. The use of the three equations in this section must explained and justified in detail. Legal provision is NOT a scientific justification – scientific articles should be based on scientific grounds, not on legal approaches.
2.4. Sampling must be described. Samples were measured from each gland separately or from both glands cumulatively?
2.4. You must express in detail the number of biological and technical samples examined for each specific test. A new table describing these must be inserted. Lack of mentioning details of biological and technical samples can cast doubts on the study.
2.5., 2.6., 2.7., 2.8., 2.9. These subsections must be inserted in a new single subsection 2.5., with sub sub-sections therein.
2.10. No ethics statement is needed for milk sampling of animals. No need to over-react, the relevant European legislation is very clear to define when discomfort is caused to experimental animals and milk sampling is not included in these guidelines. Delete altogether.
Results.
Table 1-5 present cumulative results from all samples processed. Whilst there is merit in those and they certainly contribute to the value of the work, they can be misleading, as they represent samples from a variety of management systems, health status, lactation stage etc. Therefore, they must be moved to supplementary material, as they inhibit the flow of the text reading without providing much useful information.
Discussion.
Please include comparison with results of similar efforts performed in other countries of the world.

Author Response

Reviewer 3

The introduction is unnecessarily long. Readers of the manuscript would understand the topic, hence a two-paragraph intro would suffice.

Response: Introduction was reduced as suggested by reviewer from 1137 to 569 words.

2.1. Some measures of the age of animals are important: mean±s.e. or median values will help.

Response: Reviewer suggestion was followed (mean±SD).

2.2. polyestric, please use correct terminology and also English-english language.

Response: Changed to polyestrous.

2.2. kgs is wrong

Response: Corrected.

2.2. The use of the three equations in this section must explained and justified in detail. Legal provision is NOT a scientific justification – scientific articles should be based on scientific grounds, not on legal approaches.

Response: We explained the use of the equations as suggested by the reviewer. Legal provision suggested ICAR guidelines should be taken into consideration to perform production controls. The guidelines have been set up for the first time in 1992 with the purpose of being informative more than normative and have been regularly updated since then considering the new scientific computational and measuring advances.

2.4. Sampling must be described. Samples were measured from each gland separately or from both glands cumulatively?

Response: Sampling details were provided.

2.4. You must express in detail the number of biological and technical samples examined for each specific test. A new table describing these must be inserted. Lack of mentioning details of biological and technical samples can cast doubts on the study.
2.5., 2.6., 2.7., 2.8., 2.9. These subsections must be inserted in a new single subsection 2.5., with sub sub-sections therein.

Response: Reviewer suggestion was followed and Table and subsections were added.

2.10. No ethics statement is needed for milk sampling of animals. No need to over-react, the relevant European legislation is very clear to define when discomfort is caused to experimental animals and milk sampling is not included in these guidelines. Delete altogether.

Response: Section 2.10 was removed.

Results.
Table 1-5 present cumulative results from all samples processed. Whilst there is merit in those and they certainly contribute to the value of the work, they can be misleading, as they represent samples from a variety of management systems, health status, lactation stage etc. Therefore, they must be moved to supplementary material, as they inhibit the flow of the text reading without providing much useful information.

Response: Tables were moved to Supplementary section.

Discussion.
Please include comparison with results of similar efforts performed in other countries of the world.

Response: Discussion was complemented with similar efforts carried in other countries and with additional information in regards the bases of the best fitting properties of ALISCH and PARYLDENS when compared to the rest of models.

Reviewer 4 Report

María del Amparo Martínez Martínez and colleagues investigated the method of predicting milk production based on goat milk composition and the reliability of various statistical models were evaluated. Generally, the study is of interest and convincing but the entire manuscript was poorly presented and needs a general overhaul as well as language editing. The presentation of the overall article is a bit confusing, especially the presentation part of the results. This article uses SPSS for the analysis work, but I think SAS maybe better for this manuscript because the main work in this paper is for statistical Analysis.

Lines 30: “can be evaluate and modelized…. “Grammatical errors  and I think “can” ,the tone is too positive

 Line 51: Too messy and redundant for the introduction in the preface, the author should focus on the research direction of this article and give a brief introduction.

Line153-154: These milks samples were from 2008-2015, 28 farms, how to ensure that the breeding environment of these pastures, especially the changes in the feed formula, will not affect the milk composition? As we all know, changes in formula will affect milk protein and lactose.

Lines 216: The speculate on the content represented by b0-b4 in the discussion section maybe better for this manuscript.

Lines334: Poorly stated the result. For summary of descriptive statistics table, Not a simple list of numbers. Percentile 25,50 and 75 are no need. Missing unit of measure

Line 344: It is more intuitive to use graphics to represent relevance Lines 131-132 : Provide primer sequences

Line 335 : Table 2 and Table 3 can be combined on one table, no need to list all of the number.

There are many ambiguities in the format and presentation of the results. I don’t have time to check them one by one. I hope the author can revise it and make the results intuitive

Another point, I think the length of this article may be due to the author's selection of too many models. In the discussion and preface, too many discussions are inconsistent with the results of this article. I hope the author can focus on the two models with better results in this article. Write foreword and discussion

Author Response

Reviewer 4

Comments and Suggestions for Authors

María del Amparo Martínez Martínez and colleagues investigated the method of predicting milk production based on goat milk composition and the reliability of various statistical models were evaluated. Generally, the study is of interest and convincing but the entire manuscript was poorly presented and needs a general overhaul as well as language editing. The presentation of the overall article is a bit confusing, especially the presentation part of the results. This article uses SPSS for the analysis work, but I think SAS maybe better for this manuscript because the main work in this paper is for statistical Analysis.

Response: A Cambridge ESOL examination instructor revised the manuscript to imporve readability and correct potential grammar mistakes and typos. Specifically we did not use SAS or other software as our intention was to present SPSS as a relatively easier valid alternative given many researchers are more familiarised with this software.

Lines 30: “can be evaluate and modelized…. “Grammatical errors  and I think “can” ,the tone is too positive

Resposne: Grammar and typos were checked and corrected across the body text.

 Line 51: Too messy and redundant for the introduction in the preface, the author should focus on the research direction of this article and give a brief introduction.

Response: Introduction was reduced as suggested by reviewers from 1137 to 569 words.

Line153-154: These milks samples were from 2008-2015, 28 farms, how to ensure that the breeding environment of these pastures, especially the changes in the feed formula, will not affect the milk composition? As we all know, changes in formula will affect milk protein and lactose.

Response: All farms followed permanent stabling practices, with ad libitum water, forage, and supplemental concentrate. The Murciano-Granadina breed feed formula is standardized and a further description of the detailed and analytical composition of the diet provided to the animals in the study can be consulted in Fernández Álvarez, et al. [2].

Fernández Álvarez, J.; León Jurado, J.M.; Navas González, F.J.; Iglesias Pastrana, C.; Delgado Bermejo, J.V. Optimization and Validation of a Linear Appraisal Scoring System for Milk Production-Linked Zoometric Traits in Murciano-Granadina Dairy Goats and Bucks. Applied Sciences 2020, 10, 5502.

Lines 216: The speculate on the content represented by b0-b4 in the discussion section maybe better for this manuscript.

Response: Section was transferred to discussion as suggested by the reviewer.

Lines334: Poorly stated the result. For summary of descriptive statistics table, Not a simple list of numbers. Percentile 25,50 and 75 are no need. Missing unit of measure

Response: Percentile 25, 50 and 75 were removed.

Line 344: It is more intuitive to use graphics to represent relevance

Response: A graphic representation of each best-fitting model was added as suggested y the reviewer.

Lines 131-132 : Provide primer sequences.

Response: A reference was added. Primers and genotyping conditions can be consulted in Pizarro Inostroza, et al. [3].

Line 335: Table 2 and Table 3 can be combined on one table, no need to list all of the number.

There are many ambiguities in the format and presentation of the results. I don’t have time to check them one by one. I hope the author can revise it and make the results intuitive

Another point, I think the length of this article may be due to the author's selection of too many models. In the discussion and preface, too many discussions are inconsistent with the results of this article. I hope the author can focus on the two models with better results in this article. Write foreword and discussion

Response: Tables 2 and 3 have been combined. As requested, y other reviewer they and the rest of tables except for former Table 6 have been relocated as supplementary material.: Discussion was complemented with additional information in regards to the bases of the best fitting properties of ALISCH and PARYLDENS when compared to the rest of models. Discussion was supported on the findings in our study.

Round 2

Reviewer 3 Report

Authors have answered successfully to the comments.

Author Response

Response: We thank the reviewer for his/her comments.

Reviewer 4 Report

All my questions have been solved, just one question on the figure1: needs to add numerical information (similar to the degree of fit) to show the effect of the model

Author Response

Response: Adj. R2 value was added for each graphic in Figure 1.

This manuscript is a resubmission of an earlier submission. The following is a list of the peer review reports and author responses from that submission.